# Cost-Effective, Single-Frequency GPS Network as a Tool for Landslide Monitoring

**DOI:** 10.3390/s22093526

**Published:** 2022-05-06

**Authors:** David Zuliani, Lavinia Tunini, Federico Di Traglia, Massimiliano Chersich, Davide Curone

**Affiliations:** 1National Institute of Oceanography and Applied Geophysics—OGS, 33100 Udine, Italy; ltunini@ogs.it (L.T.); fditraglia@ogs.it (F.D.T.); 2YETITMOVES S.r.l., c/o EUCENTRE, University of Pavia, 27100 Pavia, Italy; mchersich@yetitmoves.it (M.C.); dcurone@yetitmoves.it (D.C.)

**Keywords:** GPS, cost-effective, near real-time, landslide monitoring, low-cost instrumentation, Eastern Alps

## Abstract

The constant monitoring of active landslides, particularly those located in the proximity of populated areas or touristic places, is crucial for early warning and risk-management purposes. The commonly used techniques deploy expensive instrumentation that can be hardly afforded, especially by small mountain communities in which landslide events often occur repeatedly. In recent years, the scientific community, as well as the private sector, have devoted growing effort to reducing the costs of monitoring systems. In this work, we present a monitoring network based on single-frequency Global Positioning System (GPS) sensors that have been activated to monitor an active landslide in the Carnic Alps, North-Eastern Italy. The system, which was composed of 12 single-frequency GPS stations, one seismometric station coupled with a single-frequency GPS instrument for real-time monitoring, and one permanent dual-frequency GPS station located in a stable area, provided daily reports of the landslide motion to the local authorities and administration. We show that this system is a valuable, flexible, and cost-effective tool for quick landslide characterization, and has high potential to be used as a landslide early warning system in case of emergency situations.

## 1. Introduction

The study of the displacement of active landslides is crucial to define risk mitigation procedures and to implement early warning systems [1]. Displacement data can be derived from sensors installed directly on the landslide body (geotechnical monitoring), or remotely sensed with ground, airborne, or satellite devices [2]. Monitoring systems are designed to primarily satisfy two purposes: (i) understanding the extension of phenomena and their kinematics (e.g., [3]) or (ii) providing early warnings (even defining the time to failure; e.g., [4]) and identifying the variation and dynamics of landslides in real-time [5]. In the first case, the displacement data allow the evaluation of all of the dimensional and evolutionary parameters that contribute to defining the conceptual models of a landslide. It is then possible to define the state of activity and the evolution of a landslide. Additionally, the triggering causes and activation predisposition of a landslide can be evaluated. This information is useful to appropriately select the type and the sizing of any stabilization works necessary to reduce the landslide risk, as well as to support territorial planning and civil protection plans. Furthermore, understanding the size and type of movement is essential for the design of monitoring networks for alert purposes. In the second case, the monitoring network is aimed at early warning and requires continuous and near-real-time acquisition in order to define thresholds for each sensor/instrument, or mathematically treat the displacement time series in order to define the time to failure (when possible [6]).

The instruments commonly used for estimating landslide displacements are based on sensors installed in situ, such as inclinometers and extensometers, or remote sensing techniques using ground-based sensors, such as ground-based interferometric synthetic aperture radars (GBInSAR), terrestrial laser scanner (TLS), or topographic measurements captured by robotized total stations (RTS). Each of these techniques has advantages and disadvantages, but they usually require very expensive equipment. Recently, some efforts have been made to find cost-effective solutions in order to efficiently reduce instrumental costs. Among these, we can mention wireless sensor networks (e.g., [7,8]) or single-frequency Global Navigation Satellite Systems (GNSS, which includes, among others, the well-known Global Positioning System—GPS) sensors (e.g., [9,10,11,12]).

Here, a monitoring network based on single-frequency GPS sensors is presented. The objective of this work is to demonstrate that this system is innovative for the type of instrumentation used (GPS cost-effective) and address the problem of implementing efficient and cost-effective solutions dedicated to the monitoring and early warning of critical infrastructures and areas prone to hydrogeological instability. Furthermore, the system presented in this work overcomes some limitations imposed by the monitored area (highly vegetated slope, lines of sight, and distances from reference stations), and it takes into account the limited budget available. 

The object of the study is the landslide near the village of Cazzaso (Municipality of Tolmezzo, Udine, Italy) in the north-eastern Alps (Figure 1). The landslide, already reported in 1807 and reactivated in October 1851, moved the entire town by 24 m, with the destruction of some buildings (Ref. [13] and references therein). This caused the partial relocation of the town to a new location 1 km away (Cazzaso Nuova). Repeated topographic surveys and slope restoration interventions have been carried out in the area during the past and current decades (Refs. [14,15,16,17,18] and references therein). However, the instability problems persist, and they depend primarily on the groundwater conditions of the area that affect the slope stability. Therefore the implementation of a monitoring system able to track, at least in near-real-time, the landslide displacements has become a need for the local administration and authorities to timely take appropriate measures of risk containment.

The area examined in this study (Figure 1) is located in the Carnic Alps, forming part of the eastern sector of the Alpines mountain chain [19]. The slope is constituted by glacial, fluvioglacial, lacustrine, and fluvial deposits [20], lying on a calcareous–clayey bedrock. The slope where the town of Cazzaso is located has been affected by a landslide since, at least, the second half of the 19th century (Ref. [13] and references therein). The landslide has a complex movement, being an earth rotational slide that has evolved into an earth flow (based on the [21] classification), according to the descriptions reported in [13]. The landslide involves the superficial cover, which is characterized by an inhomogeneous permeability, with spatial rain drainage and the absence of water springs. The landslide movement is slow, in the order of magnitude of meters per year [13,21,22,23]. The climate in the study region is temperate with cold winters [24] (the lowest average monthly temperature is −3.7 °C in January, while the highest monthly average maximum temperature is 26.2 °C in July) and abundant precipitation in all seasons (2500–3000 mm/year); precipitation from November–December to March–April prevailingly occurs as snowfall (https://monitor.protezionecivile.fvg.it/#/stazione/180, accessed on 27 April 2022). 

**Figure 1 sensors-22-03526-f001:**
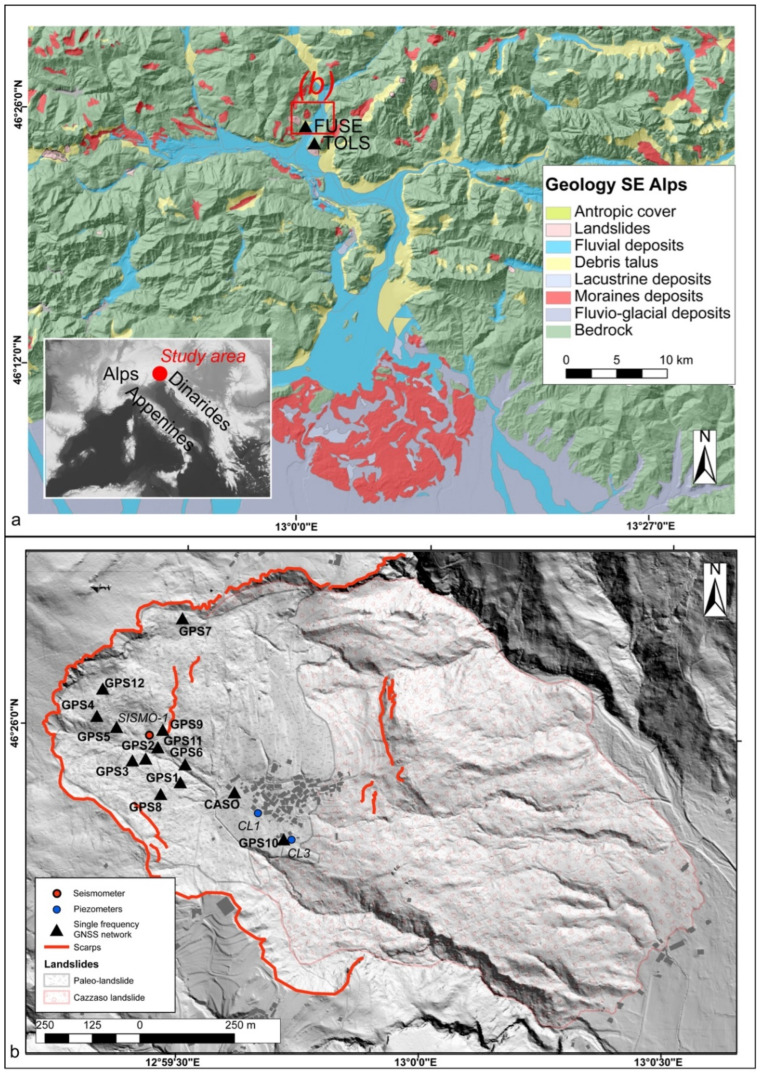
(**a**) Geological setting of the study area, located in the SE Alps, where there are large areas characterized by moraine and slope (debris talus and cones) deposits (after [25]). The locations of the FReDNet network reference GNSS stations (FUSE and TOLS) have been added. The location of the study area with respect to the Alpine chain is reported in the insert; (**b**) Distribution of the single-frequency GPS network with respect to the landslide affecting the town of Cazzaso (modified after [13] and references therein). The hillshade model was derived from a 1 × 1 m digital terrain model available through the portal of the Friuli Venezia Giulia region (http://eaglefvg.regione.fvg.it, accessed on 27 April 2022).

In this study, we present a monitoring system implemented to study the Cazzaso landslide, and we demonstrate that the single-frequency GPS sensor network is an excellent tool for understanding the kinematics of the landslide. Furthermore, it proves to be a very promising instrument to be used as a movement dynamics analysis system, potentially usable for alert purposes.

## 2. Materials and Methods

The Cazzaso landslide monitoring system (Table 1) is composed of 12 single-frequency GPS stations (from GPS1 to GP12, provided by the private company YETITMOVES, https://www.yetitmoves.it/, accessed on 31 March 2022), 1 seismometric station (SARA 24 BIT 6 channels datalogger model SL06C6, SARA accelerometer model SA10, and SARA velocimeter model PF-S-SS02) coupled with a single-frequency GNSS instrument (LZER0 designed by OGS [26]) for real-time monitoring (SISMO1), and 1 permanent dual-frequency GNSS (Trimble NetR9 model) station located in the village of Cazzaso (CASO). In the same village, one rainfall gauge is available inside the CASO monitoring station. The data from the other 2 piezometers (PA and PB) are also integrated into the monitoring system.

The GPS network has been operated since 2016 by the Seismological Research Center—CRS, belonging to the National Institute of Oceanography and Applied Geophysics—OGS. Whereas the dual-frequency station tracks both L1 and L2 carrier signals, the single-frequency devices are able to track only the L1 carrier from the GPS satellites. However, their capacity for measuring relative positions reaches centimeter-level repeatability for hourly sessions, and millimeter-level repeatability for daily sessions [27] in the post-processing mode if their data are calculated with respect to a reference station placed at a distance not exceeding 10 km [28]. It is worth noting that all of the movements measured at each station refer to the first measured position (time differencing) and that the differential approach over short baselines allows attenuating or eliminating common-mode errors (such as atmospheric biases, antenna phase center offsets and variations, etc.). On the other hand, the measurement precision at each point strongly depends on environmental conditions (possible sky impairments due to surrounding orography and vegetation). In this specific case, the sensitivity of the monitoring system has been demonstrated to be as low as 3 mm for the horizontal components and 1 cm for height when processing daily sessions. All of these characteristics, combined with their more affordable price, make the cost-effective GPS network suitable for studying the Cazzaso landslide. Furthermore, the presence of the permanent GNSS station of CASO ensures the stability of the entire monitoring system and it provides useful support to the data acquisition campaigns carried out periodically by the Municipality of Tolmezzo.

The installation of the GPS part of the monitoring system was carried out through four steps: (1) terrain recognition; (2) installation of the ground-anchored columns for each GPS site; (3) installation of the GPS antenna and receivers; and (4) activation of the monitoring network. 

The first step, terrain recognition, was aimed at selecting the best locations for the GPS stations. The recognition concerned the analysis of the geological characteristics of the terrain, the satellite visibility, the road accessibility, the availability of stable electrical power and transmission systems, and the evaluation of the absence/presence of magnetic disturbance phenomena and of obstacles that can impair the GPS signal and possibly increase the multipath. All this information has been used to select appropriate locations and monumentation types for the GPS stations. 

In the case of the area of the Cazzaso landslide, the problem was the dense tree coverage, which impeded completely satisfying all of the requirements described above. Hence, some compromises were made: good satellite visibility from 15° above the horizon and tree-free region around 10 m from the station location in order to reduce the loss-of-lock and/or to improve the exposure of the solar panels used in combination with backup batteries as power supply. A ground-anchored monument was selected to support the GPS antenna and the complementary instrumentation.

### 2.1. Instrumentation

The GPS station network consists of two different types of stations: the node station (NS), which receives the GPS signal and can transmit it only to the gateway, and the gateway station (GW), which is not only capable of retrieving its own GPS signal, but also collecting the information deriving from the other node stations located nearby and sending them all to a central server. The information between the NSs and GWs travels through an Ultra-High-Frequency (UHF) radio at 868 MHz. The data coming from the GW are addressed through a General Packet Radio Service (GPRS) modem to a remote server located in the headquarters of the CRS in Udine.

Each one of the GPS stations (Figure 2) consists of: one monumentation column, one 50 W photovoltaic panel, one GPS antenna (GPS MOBI, MBGPS-30 model, Figure 2b), one Yagi antenna for data transmission on 868 MHz, one IP66 box [29] containing the single-frequency GPS receiver UBLOX NEO M8T, the 868 MHz radio transmission system, the voltage regulator for the batteries and the photovoltaic panel, and two backup batteries providing 24Ah autonomy, and one GPRS Telit modem in the case that the station works as a gateway. Each modem is configured as a client TCP/IP and is capable of activating the socket with the central server for data transmission.

The power usage for each GPS station is 0.3 W in the case of a simple node station with an autonomy of 20 days, or 1.2 W in the case of a gateway station, with an autonomy of 5 days.

### 2.2. Managing Tool and the Calculation of Displacements

The system adopted for the configuration and control of the GPS network, as well as for data elaboration, was DISPLAYCE, which is a cost-effective solution dedicated to the monitoring and early warning of critical infrastructures and areas subject to hydro-geological instability, developed by YETITMOVES (https://www.yetitmoves.it/, accessed on 31 March 2022 [30]), and extending a former monitoring system installed by OGS [31]. DISPLAYCE was installed on a virtual server with Microsoft Windows 10. The DISPLAYCE Server Manager is an application for managing raw data acquisition from remote GPS GW stations (and attached NS stations) and converting GPS raw data to a standard Receiver INdependent EXchange (RINEX) format and storing it on a storage disk. The same application was used to process the downloaded data, to generate displacement time series for each GW and NS station, and to produce alerts and reports to be sent through email. Usually, GW and NS are standard YETITMOVES stations (which are made of a GPS U-BLOX receiver chipset), but the software is designed to directly accept RINEX files and data coming from other sources (e.g., rainfall gauges). The stations are also able to send to the server telemetry data, such as battery and solar panel powers, and the signal-to-noise ratio of the GW–NS connections and of the GPRS modem in the GW stations. DISPLAYCE software calculates the displacement of each station at regular intervals (i.e., regular sessions) of 1, 3, 6, 12, and/or 24 h, providing the different parameters of the GPS signal (signal-to-noise ratio, number of cycle slips, etc.). When a session is terminated, the server provides the average telemetry data from the same session and records them within an internal database. The data of the internal database are then used by the software to generate a report that is sent by email to the local authorities in charge of risk management (Tolmezzo Municipality and the Regional Civil Protection). A Warning system included in the code allows defining alert thresholds based on the telemetry parameters or on the recorded displacements, and written reports can be forwarded to the local authorities as well. The final user can access all of the information using a DISPLAYCE Client (again, an application designed for the Microsoft Operative System), which interrogates the remote server and visualizes the time series of the remote stations, the telemetry data, and a map with the velocities estimated from the recorded displacements. Furthermore, it allows the configuration of the report and warning generators.

The calculation of the displacements is the result of the GPS data processing, which is carried out through the widely known Double Difference (DD) technique [32,33]. The DD technique is based on the contemporary presence of at least two stations: a Rover unit located on the monitored structure, and a Master station installed on a stable structure located outside the monitored structure. The DD technique allows interring the distance between the Rover and Master, called the baseline. In our study, the Rover was represented by each one of the GPS stations located on the Cazzaso landslide (NS or GW stations), whereas the Master was the FUSE station located approximately 2.3 km away from the landslide. FUSE is a dual-frequency GPS permanent station belonging to the Friuli Regional Deformation Network (FReDNet) [34,35,36,37]—further information can be found at http://www.crs.ogs.it/frednet (accessed on 31 March 2022). Recently, the staff of CRS installed the station of TOLS (located less than 1.5 km south-east of FUSE, see Figure 1a) to ensure the functioning of the system in the case of unexpected failures at the FUSE station. As we are using single-frequency devices, we had to ensure that we relied on short baselines, i.e., less than 5 km with at least 1 h of data, and on GPS stations located roughly at the same latitude, in order to provide a significative reduction of the measurement errors induced by the troposphere and the ionosphere [28,29,30,31,32,33,34,35,36,37,38], and to ensure centimeter-level accuracy. 

## 3. Results

Since its activation at the beginning of 2016, the monitoring system has identified a strong displacement (11 cm of eastward movement) recorded by one of the GPS stations (GPS3) on 28 February 2016. The displacement occurred after an anomalous rainy event. Owing to the alert warning sent by email by the monitoring system, the local administration closed the road of access to the Cazzaso village in a timely manner, avoiding further safety-related problems.

In order to compare as many stations as possible, in Figure 3, the displacement rate measured in the interval of 1 January 2019–30 June 2021 by the different stations is shown (with the exception of the GP10, GP11, and GP12 stations for which the time series are too short), while Figure 4 presents the time series of some of the most representative stations (GPS2, GPS3, GPS9, and CASO), and cumulative rainfall data have been reported. The results allow us to point out that: (i) the stations located on the slope showed values of one or more orders of magnitude larger than those of the CASO station located in Cazzaso village; (ii) the stations moved differently within the same slope, showing velocities, as reported in Table 2; (iii) GPS3 was the station which moved the most, especially along the East component; (iv) GPS7 was moving in a different direction with respect to all of the other stations, suggesting that it was affected by a different dynamic; (v) the motion registered by the GPS stations occurred by “steps” when rainfall events occurred. These steps were generally of the order of a few tens of centimeters, but they could measure up to 0.5 m in the case of strong rainfalls, as occurred along the Eastern component of GPS3 in November 2019 (Figure 4).

From the monitoring data, it is possible to note that the stations showing greater displacements, in both the horizontal and vertical components, are those located close to the landslide crown, suggesting a differential movement of this part with respect to the rest of the slope (Figure 3).

Further in-depth analyses will allow us to understand if these movements are residual motions of the landslide that took place in the 19th century, or if they are related to more superficial phenomena, such as soil slip or soil slides, which could evolve into debris flows.

## 4. Discussion and Conclusive Remarks

The cost-effective, single-frequency GNSS network proved to be a capable tool for quick landslide characterization and, considering the good concomitance between measured displacements and recorded rainfall, a potential instrument to be used as a landslide early warning system in case of emergency situations. 

The main benefits of the network for landslide monitoring are:-Flexibility: the system can be extended over a large area; in fact, the distance of some tens of meters allows it to be used for landslides of different sizes and, therefore, volumes without requiring specific effort for the system’s adaptability;-Installable in different conditions of the slope: the system does not require special installation conditions, as is necessary for systems such as GBInSAR, TLS, or RTS. These technologies require an installation site at a stable position; however, at the same time, they do not require excessive distances (<1 km for TLS and RTS), directions of line-of-sight as parallel as possible to the direction of movement (GBInSAR), or slopes to be monitored with little vegetation (GBInSAR). On the contrary, the cost-effective network requires only one node external to the landslide and, concerning vegetation, that each node has sufficient sky visibility to be capable of retrieving the satellite signals (in the Cazzaso test site, 15° above the horizon and a tree-free region around 10 m from the station);-Cost-effective: the cost of each single-frequency receiver is reduced by about a tenth compared to that of a double frequency monitoring receiver;-Scalability: the sensor network can be made more or less “dense” according to the characteristics of the landslide;-3D monitoring: like all GNSS sensors, even single-frequency ones allow the reconstruction of 3D displacements.

This system also has limitations if compared to the GBInSAR and TLS techniques: (i) it does not allow the areal reconstruction of displacements; (ii) the system takes one hour to obtain data with centimeter-level precision. This is a limitation compared to systems, such as GBInSAR and RTS, which limits the field of applicability to all types of landslides. Despite these drawbacks, when applicable, the single-frequency GPS network implemented in this study is an effective tool for landslide monitoring, with low costs of instrumentation and maintenance.

## Figures and Tables

**Figure 2 sensors-22-03526-f002:**
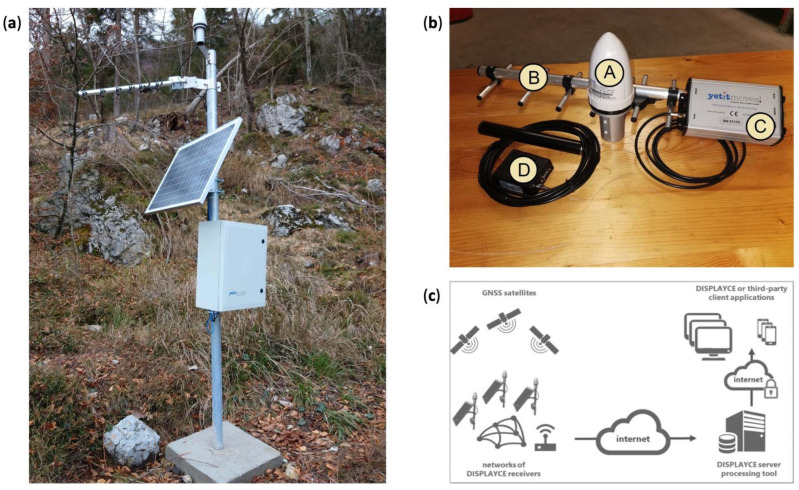
(**a**) One monumentation column anchored to the ground with DISPLAYCE instrumentation, with the solar panel and the electric box (GPS9 in this photo). The column, designed to be integral with the terrain, is made of a galvanized pipe anchored to the ground through a cemented foundation; (**b**) Instrumentation components of each station: 1 GPS antenna (A); 1 Yagi antenna for data transmission on 868 MHz (B); 1 box containing the single-frequency GPS receiver UBLOX NEO M8T, the 868 MHz radio transmission system, and the voltage regulator for the batteries and the solar panel (C); and 1 GPRS Telit modem in the case that the station works as a gateway, with the antenna (D); (**c**) Scheme of the presented monitoring system (image modified from the Displayce guide document provided by YETITMOVES company, publicly available on https://www.yetitmoves.it/wp-content/uploads/2021/11/DISPLAYCE-it.pdf, accessed on 27 April 2022).

**Figure 3 sensors-22-03526-f003:**
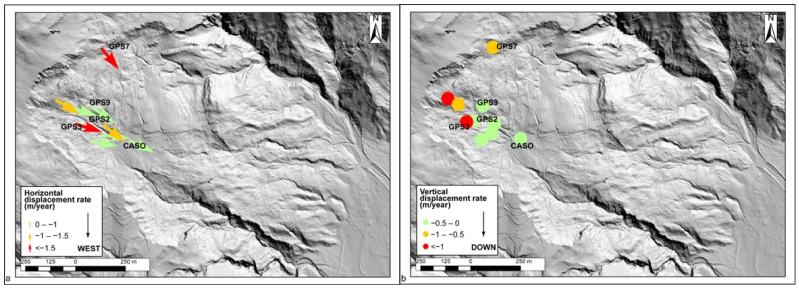
(**a**) Horizontal and (**b**) vertical displacement recorded by the cost-effective, single-frequency GPS network on the Cazzaso landslide between 1 January 2019–30 June 2021. The sensor data were used, with the exception of the data of the GP10, GP11, and GP12 stations, which are more recent and have a shorter time series than the other stations.

**Figure 4 sensors-22-03526-f004:**
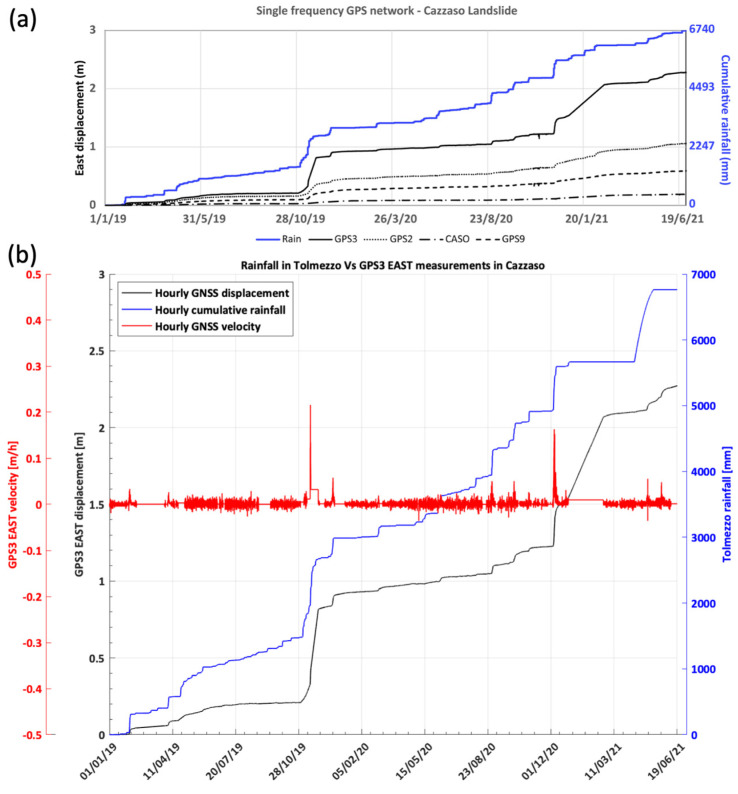
(**a**) Displacement toward the East recorded by the cost-effective, single-frequency GNSS network on the Cazzaso landslide between 1 January 2019 and 30 June 2021. The black lines with different patterns indicate the different GPS stations. The blue line indicates the cumulative measurement of rainfall recorded by the Tolmezzo rainfall gauge (near TOLS station); (**b**) Detailed plot of the GPS3 eastern displacement compared with the rainfall gauge values in Tolmezzo; the GPS3 hourly velocity (East component) is also included to highlight the fastest rain phenomena that had the greatest impact on GPS3’s movements.

**Table 1 sensors-22-03526-t001:** Instrumentation belonging to the monitoring network installed on the Cazzaso landslide.

Station	Instruments	Operating Interval
GPS1	Single-frequency GPS	2016–today
GPS2	Single-frequency GPS	2016–today
GPS3	Single-frequency GPS	2016–today
GPS4	Single-frequency GPS	2016–today
GPS5	Single-frequency GPS	2016–today
GPS6	Single-frequency GPS	2016–today
GPS7	Single-frequency GPS	2018–today
GPS8	Single-frequency GPS	2018–today
GPS9	Single-frequency GPS	2018–today
GP10	Single-frequency GPS	2020–today
GP11	Single-frequency GPS	2020–today
GP12	Single-frequency GPS	2020–today
CASO	Double-frequency GPS Rainfall gauge	2015–today
SISMO-1	Seismometer + instrument for real-time GPS data	2019–today
PA	Piezometer	2015–2019
PB	Piezometer	2015–2019

**Table 2 sensors-22-03526-t002:** Location, displacement rate in the different components, and direction of movement of the monitoring stations shown in Figure 3. FUSE and TOLS are the reference sites deployed on stable structures far from the landslide.

Station	N (°)	E (°)	Elevation (m)	Up–Down (m/year)	East (m/year)	North (m/year)	Horizontal (m/year)	Direction (°)
FUSE	46.414159	13.001142	532.076	-	-	-	-	-
TOLS	46.404378	13.014218	378.016	-	-	-	-	-
CASO	46.431848	12.993505	686.779	−0.0732	0.3304	−0.0994	0.3450	107
GPS1	46.432057	12.991639	729.649	−0.4162	2.1765	−0.3829	2.2099	100
GPS2	46.432598	12.990423	776.092	−0.4547	1.9039	−0.1665	1.9112	95
GPS3	46.432547	12.989969	789.059	−1.4300	3.7369	−0.3043	3.7493	95
GPS4	46.433579	12.988714	839.118	−1.3037	2.3120	−1.1722	2.5922	117
GPS5	46.433331	12.989399	808.717	−0.6803	1.1923	−0.7969	1.4341	124
GPS6	46.432486	12.991785	732.161	−0.1566	0.9829	−0.3507	1.0436	110
GPS7	46.435945	12.991565	804.152	−0.7251	1.3422	−1.1582	1.7728	131
GPS8	46.431764	12.990972	755.265	−0.0172	0.2335	−0.0945	0.2519	112
GPS9	46.433291	12.990992	769.645	−0.3374	0.6846	−0.3121	0.7524	115

## Data Availability

The data used in this experiment are owned by the Municipality of Tolmezzo and are available upon request.

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
