# Peer review of "Cost-Effective, Single-Frequency GPS Network as a Tool for Landslide Monitoring"

_sensors, 2022, doi:10.3390/s22093526_

Round 1

Reviewer 1 Report

Dear Authors,

I have read your excellent short note and beside few minor clarification I think it is ready to be published. I think the science content is fine of your report and the writing style is straight forward, easy to understand. I have provided an annotated PDF file where I marked few minor issues that are rather tidy the manuscript up. When you report meteorological data, it would be useful to see the numerical ranges of rainfall or intensity. Also, in similar way, when you mention dense array of sensors, I would like to see the scale you are referring to. There are also few long and complex sentences that would be better off to cut into short ones. Overall I think the manuscript after a very minor revision is ready to be accepted.

Kind regards,

Reviewer 2 Report

Title: Cost-effective, single-frequency GPS network as a tool for 2 landslide monitoring

I have gone through the paper. It is good and timely. It has good scientific contribution, but suffers from several flaws. The flaws are:

  1. Please provide full name of GPS in its first appearance. Also I do not see method and results in the abstract section.
  2. In introduction section, research gap, novelty, and objectives are not clearly mentioned.
  3. Section 2, several acronyms do not have full names.
  4. Description of experimental study area is missing. The background of landslide is missing.
  5. Please make flow chart of the methods.
  6. Accuracy of GPS is missing.
  7. Calibration process is also missing.
  8. Authors can provide google earth image of the landslide for verification.
  9. After recording Horizontal and vertical displacement, what will be its accuracy and sensitivity.
  10. Discussion and conclusion should be separated.
  11. Conclusion should contain limitation part.

Round 2

Reviewer 2 Report

The paper can be accepted